# Towards Modular Algorithm Induction

## Abstract

We present a modular neural network architecture MAIN that learns algorithms given a set of input-output examples. MAIN consists of a neural controller that interacts with a variable-length input tape and learns to compose modules together with their corresponding argument choices. Unlike previous approaches, MAIN uses a general domain-agnostic mechanism for selection of modules and their arguments. It uses a general input tape layout together with a parallel history tape to indicate most recently used locations. Finally, it uses a memoryless controller with a length-invariant self-attention based input tape encoding to allow for random access to tape locations. The MAIN architecture is trained end-to-end using reinforcement learning from a set of input-output examples. We evaluate MAIN on five algorithmic tasks and show that it can learn policies that generalizes perfectly to inputs of much longer lengths than the ones used for training.

## 1 Introduction

Many applications in artificial intelligence require the ability to learn and perform tasks that have algorithmic structure, such as learning a sequence of precise actions together with conditional decisions and branching. Therefore, an important research problem is learning to induce algorithms from input-output examples. A particular focus of research attention has been on *neural algorithm induction,* which are neural architectures for representing algorithms by including unbounded intermediate state and mechanisms for learning control flow. The neural Turing machine (Graves et al., 2014) and its successor the Differentiable Neural Computer (Graves et al., 2016) augment a recurrent network with an external memory using differentiable read/write mechanisms, and are able to learn simple algorithmic tasks such as array copying and sorting. Neural Random-Access Machines (NRAM) (Kurach et al., 2016) learn algorithms by generating a fuzzy circuit comprising of pre-defined modules together with modules for dereferencing and storing values in a fixed memory.

Existing architectures do not fully leverage a key fact that algorithmic tasks can be solved by flexibly combining the results of smaller, reusable procedures, which we call *modules*. A modular architecture aims to represent in a learning system these aspects of programming languages intended for human developers: (i) procedural abstractions that perform computations to produce outputs given some inputs and that can be reused in the algorithm multiple times; (ii) control flow constructs such as branching and loops to compose and combine intermediate results. Although there has been a recent movement toward modular neural architectures for supervised tasks (Andreas et al., 2016; Kirsch et al., 2018; Rosenbaum et al., 2018; Valkov et al., 2018; Chang et al., 2018), the design space of modular networks has not been fully explored for algorithm induction.

Towards this end, we present a new architecture MAIN (short for Modular Algorithm Induction Network) for neural algorithm induction. Our architecture consists of a neural controller that interacts with a variable-length read/write tape where inputs, outputs, and intermediate values are stored. Each module is a small computational procedure that reads from and writes to a small fixed number of tape cells (a given fixed set of modules are specified in advance). At each time step, the controller selects a module to use together with the tape location of the module's input arguments and the write location of the module output. This architecture is trained end-to-end using reinforcement learning.

A comparison of architectural design choices in MAIN with those in recent neural program induction approaches NTM (Graves et al., 2014), LSA (Zaremba et al., 2016), NRAM (Kurach et al., 2016), NGPU (Kaiser & Sutskever, 2016), CRL (Chang et al., 2018), and DNC (Graves et al., 2016) is presented in Table 1. Unlike previous architectures, MAIN allows for learning to compose modules

| | NTM | LSA | NRAM | NGPU | CRL | DNC | MAIN |
|---|---|---|---|---|---|---|---|
| Modules | ✗ | ✗ | ✓ | ✗ | ✓ | ✗ | ✓ |
| - General module selection | | | ✗ | | ✓ | | ✓ |
| - General argument selection | | | ✓ | | ✗ | | ✓ |
| Read/write history | ✗ | ✓ | ✗ | ✗ | ✗ | ✓ | ✓ |
| Random access | ✓ | ✗ | ✓ | ✗ | ✓ | ✓ | ✓ |
| Memoryless controller | ✗ | ✗ | ✓ | ✓ | ✗ | ✗ | ✓ |
| Encoder type | Attn | FF | FF | CNN | RNN | Attn | CNN + Attn |

Table 1: Comparison of design choices used in MAIN to recent algorithm induction approaches.

together with the corresponding argument values for the chosen modules using a general domain-agnostic mechanism for module and argument choices. It uses a generic linear tape layout together with a parallel history tape of landmark symbols that indicate the most recently read and written cells. Finally, the architecture allows for random access of the memory cells and uses a memoryless controller with a length-invariant self-attention based encoding of input tape contents.

There are two key design choices in MAIN that we found crucial for good performance. The first is in the *representation of the previous history of the computation.* Like previous work (Graves et al., 2016), we find that providing history to the model is an important source of information. We introduce a simple but effective discrete representation, introducing a parallel *history tape* of landmark symbols that indicate the most recently read and written cells. This helps the model to learn common patterns of control flow, such as maps and reduce operations over the tape. The second choice is in the *encoder architecture* of the abstracted tape view with the controller. We find that using an attention-based encoder performs much better than a recurrent network based encoding within the controller, which has been employed by previous architectures for neural algorithm induction.

We evaluate our architecture MAIN on five algorithmic tasks including array copying, reversing, increment, filter-even, and multi-digit add. MAIN can learn these tasks in an end-to-end manner using input-output examples and is also able to generalize to longer input lengths. We observe that both parallel tape history and length-invariant input tape encoding based on self-attention are necessary for the architecture to learn these tasks. Moreover, for tasks such as filter-even and multi-digit add that require the controller to perform some computations, we observe that abstracting the tape contents to only certain landmark positions can help it learn the corresponding algorithms.

## 2 RELATED WORK

**Learning to Compose and Modular Networks:** Compositional Recursive Learner (CRL) (Chang et al., 2018) is a framework for learning algorithmic procedures for composing representation transformations, where both the transformations and their compositions are simultaneously learnt from a sparse supervision. The controller that learns to compose transformation is trained using reinforcement learning, whereas the transformations themselves are trained using supervised learning. The CRL architecture supports two forms of transformations: reducers and translators. Because of specialized transformations, CRL imposes a restriction on always selecting either full tape or three consecutive input tokens as arguments (that result in 1 resulting token) for the transformations. In contrast, our architecture allows the controller to select arbitrary locations on the input tape to select as module arguments and also the write locations for their outputs.

There has also been some recent related work on learning Modular Networks (Kirsch et al., 2018) and Routing Networks (Rosenbaum et al., 2018). In routing networks, a router learns to select a sequence of function blocks to compose given an input, and is trained using reinforcement learning. Kirsch et al. (2018) use a probabilistic model to represent the module choice as a latent variable and use Expectation Maximization (EM) to learn the module parameters and the decomposition choice in an end-to-end manner by maximizing a variational lower bound of the likelihood. Both of these approaches learn modular composition of functions with the aim of reusability and better generalization across multiple tasks. Our architecture, in contrast, is designed for learning algorithms where the modules are pre-specified but the task of the controller agent is to learn to compose those modules to achieve desired input-output behavior.

**Neural Program Induction and Synthesis:** Several neural architectures have been proposed recently to learn algorithmic tasks. Neural Turing Machine (NTM) (Graves et al., 2014) extends an LSTM controller with an external memory together with differentiable read-write mechanism. Differentiable Neural Computer (DNC) (Graves et al., 2016), a successor to NTM, added capabilities of freeing up unused memory for processing long sequences and a temporal link matrix for better tracking the write order. Zaremba et al. (2016) propose an architecture with an RNN controller that learns to navigate a structured input grid by selecting move actions and outputs tokens on an output grid. Unlike our architecture, which separates control from compute, these architectures require the controller to learn both the control structure as well as the desired computations.

Neural RAM (NRAM) (Kurach et al., 2016) uses a neural controller that learns to produce a fuzzy circuit consisting of a fixed number of modules. The controller learns to wire the modules with inputs coming from a fixed set of registers or intermediate outputs, and also learns to write output back to registers and memory tape. There are three key differences between Neural RAM and our architecture. First, NRAM uses a fixed sequence of 14 modules (with an option to repeat the whole sequence multiple times). In contrast, our controller learns to select appropriate module at each time step. Second, since registers and memory tape in NRAM architecture store distributions over integers, the modules also need to be differentiable to compute output over such inputs. Our architecture, on the other hand, does not require the modules to be differentiable and also writes discrete values on the input tape. Finally, our architecture uses a parallel history tape of recent read and write positions for accessing the input tape unlike pointer dereferencing in NRAM.

We summarize the characteristics of these architectures, comparing them to our work in Table 1. To understand the rows of this table, "Modules" means whether computation is performed by composable modules. "General module selection" means the modules can be selected in any order at any point in the computation (NRAM enforces a fixed ordering). "General argument selection" means module arguments can come from anywhere in memory without restriction (CRL enforces inputs be locally adjacent). "Read/write history" means that history of previous read and write locations is provided as context to the controller. "Random access" means the architecture can look up arbitrary memory contents and agregate information across memory regions, rather than be restricted by relative head movements (LSA). "Memoryless controller" means the architecture does not require the controller to have hidden memory over time, i.e. the controller is not recurrent, and only uses external memory for context. Finally, the "Encoder type" indicates how memory is consumed by the controller, feed-forward (FF), convolutional (CNN), recurrent (RNN), or attention-based (Attn).

There is also a growing interest in using neural models to generate symbolic programs as output. RobustFill (Devlin et al., 2017) trains an attention based encoder-decoder model that generates a program given a set of input-output examples. Deepcoder (Balog et al., 2017) learns a probability distribution over a set of functions in a DSL to guide an enumerative search. Chen et al. (2019) also use an encoder-decoder based approach to synthesize Karel programs (Bunel et al., 2018), but in addition also execute the partially decoded program to guide the decoder. In contrast, our architecture uses a more general mechanism to read and write on the variable length input tape, and uses reinforcement learning to learn to compose the desired modules for each individual task.

## 3 ARCHITECTURE

We describe MAIN, an architecture which can express arbitrary modular Turing machines. Our architecture has three parts: the *memory* which stores the state of the computation, the *modules* which update the tape, and the *controller* which chooses a module $M_i$ to execute at each stage of the computation as illustrated in Figures 3. Once $M_i$ is selected out of the module set (4 modules depicted), and the read/write locations on the tape are chosen, the module sees only the read inputs, and modifies the tape only at the write position. Each module is a user provided function (which could in principle also be learned). In our experiments we picked modules of two arguments and one output, but our architecture is agnostic to the number of read/write heads as shown in Figure 1.

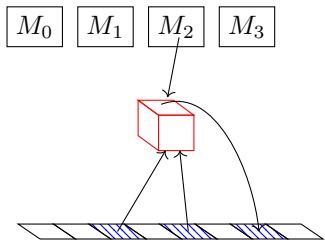

Figure 1: Module interaction with memory

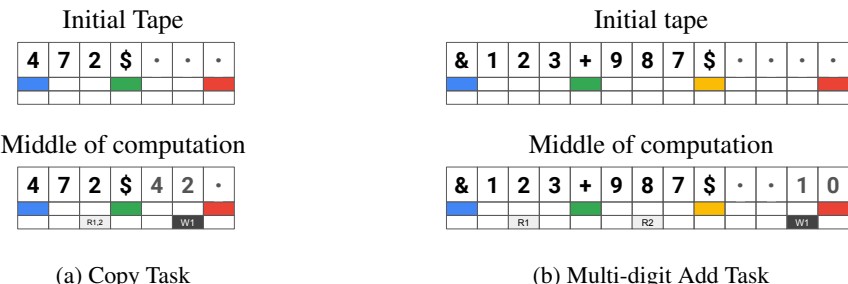

Figure 2: Example initial and intermediate tapes for (a) Copy and (b) Multi-digit add task.

The **memory** is a finite tape of discrete tokens/symbols. Formally, let $s_0, s_1, \ldots, s_T$ be memory states at each computation step $t = 0, \ldots, T$. Each $s_t$ is itself an array of tokens, indexed as $s_t[0], \ldots, s_t[L-1]$. At the start of the computation, $s_0$ is initialized with the program input, and necessary empty space to perform intermediate computation and write the output. The memory length $L$ is dynamically set based on the size of the given input, so every neural architecture component that depends on $L$ needs to be length invariant.

The memory contains *landmark tokens*, which are are task-specific tokens, e.g. '$', '+', '&', that provide positional information. For example, these tokens indicate where the input starts, where the output should be written, etc. In our experiments we found that the controller would often overwrite the landmark tokens during training. As a workaround, we provide the positions of landmark tokens as immutable metadata. These are the lambdas $\lambda^{(1)}, \lambda^{(2)}, \ldots$ inside $\sigma_t$ shown above. Additionally, we place Start-Of-Tape and End-Of-Tape landmarks at the first and last tape positions.

The initial tape $s_0$ contains the input and additional space for scratch work. At the end, the output of computation will be read from designated regions of the tape, called *target locations*, which are initialized with the empty token '.'. Note that the architecture is free to overwrite any position during the course of computation, including important input tokens. When scoring the final tape $s_T$, we only look at designated target positions, designated by the '.' token. The initial tape configurations for Copy and Multi-digit Add Tasks is shown in Figure 2. The tape configuration in addition to the input tape also consists of landmark positions as well as the read/write heads history.

The **modules** $M = \{m_1, \ldots, m_k\}$ are functions that read a narrow portion of the memory, and compute new values that are stored in memory. Each module $m_i$ is a function of $R$ arguments, and outputs a vector of size $W$, i.e. number of read/write heads corresponds to the number of inputs and outputs of the modules. During the computation, each input and output of the modules will be a single cell on the memory tape. Some example modules are the *maximum* module which returns the maximum of two inputs, and the *sum* module which returns the sum of two inputs mod base $B$.

The **controller** is a policy over actions that specifies a module-tape interaction. The controller cannot directly modify memory. Instead it selects a module and the locations on the tape the module will read from and write to. Specifically, the controller defines a distribution $\pi(H^{(r,1)}, \ldots, H^{(r,R)}, H^{(w,1)}, \ldots, H^{(w,W)}, M \mid c_t)$, where $H^{(\cdots)}$ are random variables over tape positions (support is $\{0, \ldots, L-1\}$); one for each head. The variables $H^{(r,1)}, \ldots, H^{(r,R)}$ select read-head locations, and $H^{(w,1)}, \ldots, H^{(w,W)}$ select write-head locations. The random variable $M$ is one of the modules. Finally $c_t$ is the *context*, which contains the current state of memory and information about the computation history. We will define this in more detail shortly.

The controller defines an end-to-end computation as follows. At each step $t$ of the computation, let the current state of the memory by $s_t$. To choose the next module and the locations of the read and write heads, we sample from the controller. This produces a module choice $m_t$, and locations $h_t^{(r,1)}, \ldots, h_t^{(r,R)}$ for the read heads, and $h_t^{(w,1)}, \ldots, h_t^{(w,W)}$ for the write heads. Because the controller is free to choose any tape location for each read and write head independently, the memory is random access. The tape is updated by calling the module $m_t$, with the tape contents under the read heads as input, and then writing to the position specified by the write heads. More formally,

$$s_{t+1}[h_t^{(w,j)}] := m(s_t[h_t^{(r,1)}], \ldots, s_t[h_t^{(r,R)}])[j] \qquad \forall\, j \in \{1, \ldots, W\}$$

$$s_{t+1}[i] := s_t[i] \qquad \forall\, i \notin \{h_t^{(w,1)}, \ldots, h_t^{(w,W)}\}.$$

The selected write indices are updated with module's output, and other tape elements are unchanged.

The **context** $c_t$ is the input to the controller, which represents both the current tape contents $s_t$ and an *action history* that represents information about the previous computation. The action history consists of two parts:

- *Fixed-size part:* Contains the module choice and head locations chosen at the previous time step.
  - *Module choice:* $m_{t-1}$
  - *Tape values underneath heads:* $s_t[h_{t-1}^{(r,1)}], \ldots, s_t[h_{t-1}^{(r,R)}], s_t[h_{t-1}^{(w,1)}], \ldots, s_t[h_{t-1}^{(w,W)}]$.
- *Variable-size part:* One-hot encoding of $\{h_{t-1}^{(\cdots)}\}$, the read and write head locations at the previous step of the computation.

The tape values underneath the heads are technically redundant information, because the controller can use $s_t$ combined with the head locations from the variable-sized part to lookup the corresponding tape values. However, we found that the controller had a hard time doing this in our experiments, and providing the head values as auxiliary input to the controller proved helpful.

Formally, the context is a tuple $c_t = (\xi_t, \sigma_t)$, where $\xi_t$ is a fixed sized encoding (does not depend on the tape length $L$), and $\sigma_t$ is a variable sized encoding (depends on the tape length $L$). First, $\xi_t$ describes the current tape values underneath the tape heads chosen at the previous time step:

$$\xi_t = \langle s_t[h_{t-1}^{(r,1)}], \ldots, s_t[h_{t-1}^{(r,R)}], s_t[h_{t-1}^{(w,1)}], \ldots, s_t[h_{t-1}^{(w,W)}], m_{t-1}\rangle.$$

where $m_{t-1}$ is the previous module choice. Second, the variable sized encoding $\sigma$ is a matrix concatenation that provides complete information about $s_t$:

$$
\sigma_t = \begin{pmatrix}
\mathbb{1}_V\{s_t[0]\} & \mathbb{1}_V\{s_t[1]\} & \cdots & \mathbb{1}_V\{s_t[L-1]\} \\
\hline
& \mathbb{1}_L\{\lambda^{(1)}\} & & \\
& \mathbb{1}_L\{\lambda^{(2)}\} & & \\
& \vdots & & \\
\hline
& \mathbb{1}_L\{h_{t-1}^{(r,1)}\} & & \\
& \vdots & & \\
& \mathbb{1}_L\{h_{t-1}^{(w,1)}\} & & \\
& \vdots & &
\end{pmatrix}.
$$

We can think of $\sigma_t$ as a stack of binary channels, each a row-vector of length $L$. The one-hot function $\mathbb{1}_D\{d\}$ produces a vector of length $D$ which is filled with 0s, and 1 at position $d$. Each $\mathbb{1}_V\{s_t[\ell]\}$ is a one-hot vector of token $s_t[\ell]$. Their horizontal concatenation produces a channel for each token (indicates its presence or absense). The $\mathbb{1}_L\{\lambda^{(i)}\}$ and $\mathbb{1}_L\{h_{t-1}^{(\cdots)}\}$ channels indicate the positions of landmarks and heads respectively ($\lambda^{(i)}$ are fixed throughout the episode and not indexed by $t$). Note that at $t = 0$ no actions have been taken, so the previous-action encodings are all 0s.

Now we describe the **controller architecture**. Given the context $c_t$, the controller begins with a sequence encoder for the variable length encoding $\sigma_t$. We pass $\sigma_t$ through two 1D convolutional layers (along the $L$ dimension) with filter width 3 and stride 1. This is the only aspect of our encoder that provides awareness of the local ordering of cells on the tape.

After that, we explore two different seq2fixed encoders:

- *RNN encoder:* BiLSTM produces fixed length embedding (concat of embedding for each direction) which is fed into the controller.
- *Attention encoder:* A context-independent query set is learned (fixed during evaluation). Queries are fed into attention over the tape, and the resulting weighted sum over tape values is a fixed length embedding that is fed into the controller.

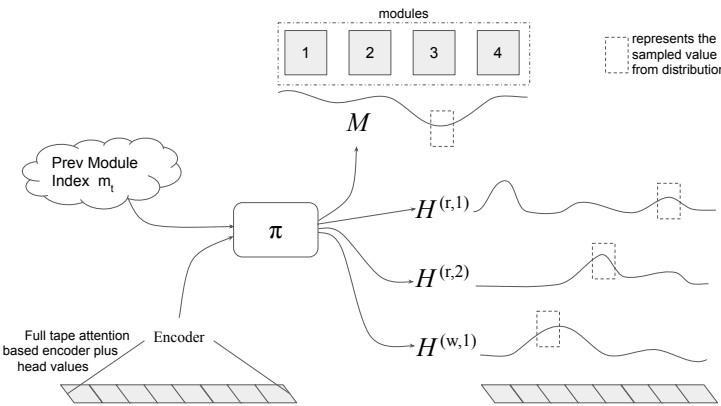

Figure 3: Controller interaction with memory and module. Controller consumes memory state from the previous timestep $t$ using an encoder, and outputs a module $M_i$ from $\{M_1, \ldots, M_k\}$, and attention weights over memory positions that determine the read and write positions for that module.

Next, we pass the resulting fixed-sized embedding for $\sigma_t$ along with $\xi_t$ into a feed forward network which outputs attention queries for the read/write head actions, and logits for the module selection action. Logits for the read/write head actions are produced with dot-product attention over $\sigma_t$ (independent attention head for each read/write head).

**Learning the Controller.** We frame our setup in the reinforcement learning paradigm, where the controller is the agent, and other components (memory and modules) are part of the environment. From the perspective of algorithm induction, only the program input and target outputs are external, while everything in MAIN is part of the black box that performs the computation. The controller is trained end-to-end on all its actions with Impala (Espeholt et al., 2018), a distributed variant of REINFORCE. Simultaneously learning to halt with RL, while learning what computation to perform, proved to be unstable in our experiments. We removed the additional complication of learning when to halt by providing a halting oracle. The oracle is given the correct output, and immediately ends the episode if the answer is in memory. The oracle is used in evaluation as well.

## 4 EXPERIMENTS

We now present an empirical evaluation of our architecture MAIN in order to establish that (i) it can learn to perform algorithmic tasks, (ii) attention is important for length generalization, and (iii) separating control flow and data flow through limited view helps in learning.

**Tasks:** We consider five algorithmic tasks with six pre-defined modules. All but the Multi-Digit Addition task are given the same module pool. We found that Multi-Digit Addition was more difficult to learn, and to reduce the action space we cut down the module pool to only ones needed to perform the computation. For simplicity, we make all the modules have the same number of inputs and outputs. Specifically in our experiments there are two read-heads and one write head. Modules which naturally read less than 2 inputs ignore their additional inputs. We consider the following six modules: Identity, Increment, Max, Sum, SumInc, and Reset. The semantics of the modules is presented in Appendix A.1. We consider the following tasks.

- *Copy*: Given an array of base 10 digits in the memory tape and a pointer to the destination, the task is to copy all elements from the array to the destination (e.g. see Figure 2(a)). We provide landmarks to start of the input and start of the output location. For an input array of length $n$, the output would be also of length $n$, and so the total memory tape size would be $2n + 1$, including 1 separation token. Modules: Reset, Identity, Increment, Max, Sum
- *Reverse*: Same tape size as Copy task, with the goal of writing the input digits in reverse order.
- *Increment*: Same tape size as Copy task, with the goal of writing the result of adding 1 to each digit of the input array (modulo base) to the destination.

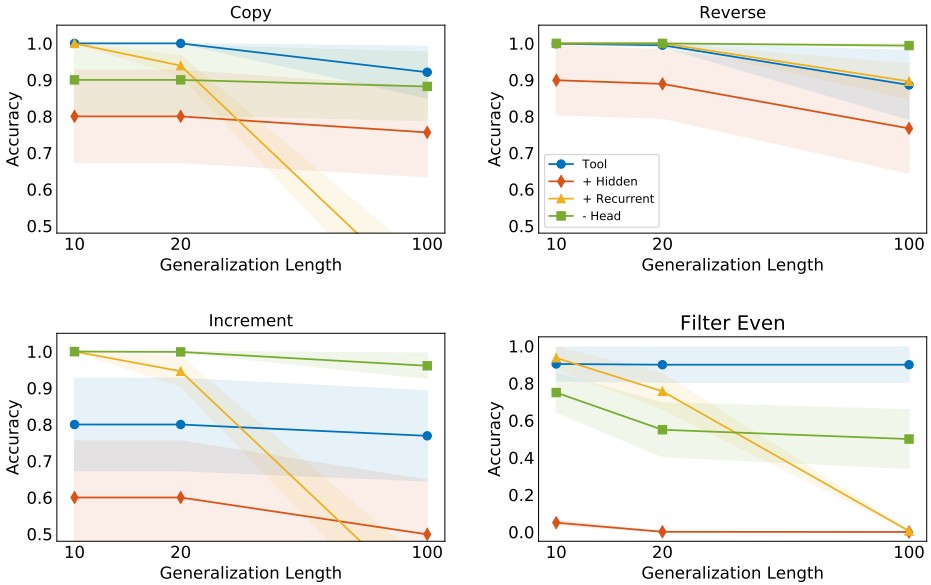

Figure 4: The average success rate and the variance of MAIN for different algorithmic tasks for different input generalization lengths of 10, 20, and 100.

- *Filter Even*: Given an array of base 16 digits in the tape, the goal is to output a sequence containing only the even-valued digits in the same order. Modules: Reset, Identity, Increment, Max, Sum
- *Multi-Digit Addition*: Given two arrays of base 10 digits separated by '+', the goal is to output the sum of the integers (denoted by input arrays) as a sequence of digits. Modules: Sum, SumInc

## 4.1 EXPERIMENTAL SETUP

We train the controller with Impala, a distributed asynchronous algorithm. We used 50 data collection workers to sample episodes from the most recent policy. There is one training worker which queues up episodes sent by the data collectors into training batches. We train until 30M timesteps across all episodes. We use a curriculum over task difficulty, which essentially corresponds to the input length. For each episode, an input-output pair is sampled from the task generator given a difficulty level, and the difficulty level is sampled uniformly in range $[1, C]$, where $C$ is the maximum curriculum setting. We vary $C$ from 2 to 10 with a linear schedule starting at 1M and ending at 18M. During training, each data collector generates new task inputs on the fly.

For evaluation, we precompute datasets of 100 test inputs for generalization with larger input lengths 10, 20, and 100 sampled from the same data generator. We run evaluation repeatedly and concurrently with training, and take the highest observed success rate as the final metric. For a given input, we compute success as whether the controller controller produced exactly the desired output. We take the average success across the 100 evaluation inputs as the success rate. Because we run each experiment 10 times, we can estimate the variance of success rate due to random weight initialization, randomly sampling from the policy, and stochastic effects of asynchronous training. We report average success rate (across the 10 trials) and the empirical standard deviation.

## 4.2 RESULTS AND ABLATIONS

Table 2 presents the experiment results of evaluating MAIN on the five algorithmic tasks with different ablation choices for the generalization input length of 100. For each task, we report the number of runs out of 10 that achieved 100% success on the test set of 100 evaluation inputs (each of length 100). Additionally, the average of success rates together with their variance for four of the tasks for different input generalization lengths of 10, 20, and 100 is shown in Figure 4.

|  | Copy | Reverse | Increment | Filter Even | Multi-Digit Add |
|---|---|---|---|---|---|
| Attention Encoder | **7** | 7 | 5 | **9** | 0 |
| – No Tape Values | 3 | 6 | 1 | 0 | **1** |
| – No Action History | 0 | 0 | 0 | 0 | 0 |
| – No Action History Tape Values | **7** | **8** | **7** | 5 | 0 |
| Recurrent Encoder | 0 | 3 | 0 | 0 | 0 |

Table 2: Ablations and variability of success. For each task, we report number of runs out of 10 that achieve 100% success on length 100 inputs. Each row is a different setting of our architecture.

As shown in Table 2, MAIN can learn to generalize perfectly for inputs of length 100 when trained on inputs with length up to 10 only. For the copy, reverse, increment, and filter-even tasks, MAIN can generalize in majority of runs out of 10. The multi-digit addition task is particularly challenging in our architectural setting. For learning this task, the controller first needs to learn to select appropriate individual digits to be added for each timestep. In addition, it also needs to learn to use the digits and module choices selected in the previous timestep to decide whether to add a carry or not while computing the addition. Remarkably, MAIN was able to learn one such controller.

Now we evaluate whether the special architectural features of MAIN are necessary for good performance. First, we evaluate whether the controller needs to observe the values on the tape, by considering an ablation (labeled "No Tape Values" in the table) which removes the top third of $\sigma_t$ (i.e. removes the tape values $s_t$). It may then seem like the controller cannot do anything, but it still has access to the action history metadata and values under the previously placed read/write heads. Notably, we found that without tape values, the controller performance goes down, except in Multi-Digit Addition, which is the only setting in which could generalize to length 100. Since this task requires the controller to compute whether a carry bit should be used for adding the intermediate result for two digits, hiding the tape contents and only providing the head values constrains the space of possible argument choices for modules and helps the controller to learn desired computation.

Next, we evaluate the usefulness of action history, by considering an ablation (labeled "No Action History" in the table) which removes the fixed context $\xi_t$ and the bottom third of $\sigma_t$, so that the controller does not have information about previous actions. As expected, these architectures do not generalize, achieving generalization accuracy of $0$. The reason is that without the action history, the controller does not have a mechanism to remember which part of the computation it is at currently, and therefore cannot learn iterative computations, which is required by all of our tasks.

Recall that $\xi_t$ contains the tape values underneath the previous read-write locations, even though the controller could infer this from other context. Next, we evaluate whether this is helpful by removing this in the ablation labeled "No Action History Tape Values". On most tasks, the performance is comparable to the full model, except on filter-even in which this ablation is slightly better.

Finally, we evaluate whether our attention-based encoder could be replaced with a simpler recurrent controller, as has been used in previous work (Chang et al., 2018). For shorter inputs of length 10, the recurrent encoder achieves perfect average evaluation accuracy for the simpler tasks such as copy, reverse, and increment (Figure 4). However, when evaluated on longer input lengths, its performance degrades significantly. For filter-even and multi-digit addition tasks, the generalization accuracy of the recurrent encoder goes to almost $0$ for length 100. On the other hand, attention based tape encoder always result in significantly higher generalization accuracies across all the tasks.

## 5    CONCLUSION

We presented a new neural architecture MAIN that learns algorithmic tasks from input-output examples. MAIN uses a neural controller that interacts with a variable-length input tape to learn to compose modules. At each time step, the controller chooses which module to use together with the corresponding tape locations for module arguments and for writing the module output back to the tape. This architecture is trained end-to-end using reinforcement learning and we show that it can learn several algorithms successfully that generalize perfectly to inputs of much longer length (100) than the ones used for training (up to 10).

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

## A  APPENDIX

### A.1  SEMANTICS OF MODULES

The semantics of the modules we consider are as follows.

**Modules:**

- $\text{Reset}(\_, \_) \to$ '.'.
- $\text{Identity}(x, \_) \to x$.
- $\text{Increment}(x, \_) \to \text{char}(\text{int}(x) + 1)$ if isnumeric$(x)$ else '.'.
- $\text{Max}(x, y) \to \max(x, y)$.
- $\text{Sum}(x, y) \to \text{char}((\text{int}(x) + \text{int}(y))\%B)$ if isnumeric$(x)$ and isnumeric$(y)$ else '0'.
- $\text{SumInc}(x, y) \to \text{char}((\text{int}(x) + \text{int}(y) + 1)\%B)$ if isnumeric$(x)$ and isnumeric$(y)$ else '0'.

