# OpenReview forum: "Towards Modular Algorithm Induction"
_ICLR.cc/2020/Conference — Reject_

### Official Review · AnonReviewer3 · 2019-10-20
**Official Blind Review #3**

**Rating:** 1

**Review:**

This paper proposes a novel modular neural architecture for algorithm induction. The modules are fixed and a controller policy is learned which outputs a distribution over modules and input/output locations on a memory tape. An oracle (which knows the correct answer) is necessary to decide when to stop computing. The controller is trained by a variant of REINFORCE.

My main concern with this paper is that there is zero experimental comparison against previous neural program induction approaches. It makes it difficult to evaluate whether the specifics of the proposed architecture actually are advantageous (albeit the authors argue so in table 1, it is not clear that these differences translate in better learning).

I find it rather disappointing that the controller needs to be stopped using an oracle which knows the right answer, making the practical use of such an architecture essentially infeasible in practice. It is also disappointing that a different subset of predefined modules is chosen for each task. I understand it helps to not have unnecessary modules in the module set, but it seems to be another weakness of the approach, and there was no evaluation against using the same full set for all tasks. Also, it is not clear that the chosen set of modules is sufficiently  general and universal, which  would be necessary to scale this approach to learn arbitrary programs.

Since the controller is trained by REINFORCE, one could be concerned that the gradient estimator has high variance (compared to methods based on soft-attention where one can backprop all the way through a sequence of actions). It would thus have been good to verify how sample complexity worsens as the complexity of the task  scales up, but there is no such analysis.

Considering the above issues, I suggest to reject that submission.

Minor:

Page 4, 2nd par, the text refers to sigma_t "shown above", which has not yet been introduced (comes in the middle of page 5).

Page 7, 2nd par, "linear schedule starting at 1M and ending at 18M" needs to be clarified. 1M what?

Page 8, "generalize perfectly": it does not seem to be the case since none of the experiments have led to 10/10 successes.  Are the "runs" mentioned training runs?


**Experience Assessment:**

I have published in this field for several years.

**Review Assessment: Checking Correctness Of Derivations And Theory:**

N/A

**Review Assessment: Checking Correctness Of Experiments:**

I carefully checked the experiments.

**Review Assessment: Thoroughness In Paper Reading:**

I read the paper thoroughly.

---

### Official Review · AnonReviewer1 · 2019-10-22
**Official Blind Review #1**

**Rating:** 1

**Review:**

This paper proposes Modular Algorithm Induction Network (MAIN) that learns algorithms given input-output examples. MAIN is equipped with several components that make it perform better than baselines, but probably the most important part is its use of modules to break-down algorithmic tasks into simpler problems. MAIN is learned end-to-end using reinforcement learning and demonstrated to perform well in several tasks.

The paper provides a good comparison between the proposed architectures and other related works. The idea to compose modules is reasonable. Nevertheless, I find this paper somewhat incomplete and not ready for publication. The most critical part is its lack of experimental validation. Table 2 presents a bunch of success rates without comparison to other methods. Unless being an expert or practitioner having the experience to implement baselines, it is hard to assess how good the proposed algorithm is. The authors provide some ablations and some of them may roughly correspond to the baselines, but the most import part - introducing modules and a mechanism to compose them actually helps - is not demonstrated in the paper at all. Hence I think the experiment section demonstrated nothing but a proof-of-concept (the proposed algorithm seems to work).

The main section describing the algorithm is not very comprehensive. I could imagine how difficult it would be to reproduce the results in the paper. It would be better to specify more detailed architectural choices.

Overall, I agree with the main point of the paper but vote for the rejection for now, and the paper needs a major revision on its experimental validation and presentation.

**Experience Assessment:**

I do not know much about this area.

**Review Assessment: Checking Correctness Of Derivations And Theory:**

N/A

**Review Assessment: Checking Correctness Of Experiments:**

I assessed the sensibility of the experiments.

**Review Assessment: Thoroughness In Paper Reading:**

I read the paper at least twice and used my best judgement in assessing the paper.

---

### Official Review · AnonReviewer2 · 2019-11-13
**Official Blind Review #2**

**Rating:** 1

**Review:**

This paper proposes a new variation of modular neural networks. Specifically, they proposed a new architecture for the neural controller that selects which module to use given the current state and history computations, and evaluated the architectures for five symbolic tasks. I'm leaning towards rejection because:
1. The story is confusing. Instead of framing algorithm induction as a program synthesis task, the paper told the story from the modular networks's view. This is quite confusing as most of work in this line of research usually assumes the modules are neural networks and are learned, while this paper uses user provided modules. This should then be framed as doing program synthesis with pre-specified primitive functions. This work hence is quite close to NPI [1]. A RL version of NPI is also recently proposed [2].
2. Lack of novelty. Continuing from the previous point, it seems it is not easy to distinguish the main idea of this work and the work of NPI. There are two differences. There's an architectural difference about how one deals with history. NPI uses a LSTM and this work uses the last time step's computation configuration as input to a FNN. It is unclear why one is better than the other? Such modification is also lack of motivation, unexplained in the paper. Another difference is about how one designs the tasks/environments. This work uses a working tape while NPI uses programming traces.
3. Lack of baseline comparisons. In the experiment section, no previous work are evaluated and compared to the proposed method, leaving one unknown how well the proposed method compares with previously known work.

That being said, there are still some merits in the paper. The currently known work for training NPI with RL [2] does not support argument predictions, while this work does. I do not know if this is due to the interface/environment that's designed for the tasks in the paper, but I recommend the authors to rewrite the story and clarify these differences more clearly. Also how general is this environment? For harder programming tasks, can one still uses tape? How scalable is this?

Minor point
- How do you use FNN to produce all actions of the controller? Do you use a different heads for different action output? Or do you use an autoregressive architecture? Have you tried both and compared?

[1] Neural Programmer-Interpreters.
[2] Learning Compositional Neural Programs with Recursive Tree Search and Planning.

**Experience Assessment:**

I have read many papers in this area.

**Review Assessment: Checking Correctness Of Derivations And Theory:**

N/A

**Review Assessment: Checking Correctness Of Experiments:**

I carefully checked the experiments.

**Review Assessment: Thoroughness In Paper Reading:**

I read the paper thoroughly.

---

### Author Response · Authors · 2019-11-13
**Thank you for the reviews**

Thanks to all of the reviewers for your time and for the very helpful comments. We will take them into account when revising this paper.

---

### Decision · Program_Chairs · 2019-12-19

**Decision:**

Reject

**Comment:**

The reviewers all agreed that although there is a sensible idea here, the method and presentation need a lot of work, especially their treatment of related methods.